# Novel Magnetically Charged Grafts for Vascular Repair: Process Optimization, Mechanical Characterization and In Vitro Validation

**DOI:** 10.3390/polym17131877

**Published:** 2025-07-05

**Authors:** Iriczalli Cruz-Maya, Roberto De Santis, Luciano Lanotte, Vincenzo Guarino

**Affiliations:** 1Institute of Polymers, Composites and Biomaterials (IPCB), National Research Council of Italy, Mostra d’Oltremare Pad. 20, V.le J. F. Kennedy 54, 80125 Naples, Italy; 2Department of Physics “E. Pancini”, University of Naples Federico II, Piazzale V. Tecchio 80, 80125 Naples, Italy

**Keywords:** electrospinning, magnetic particles, mechanical response, polycarbonate urethanes, in vitro studies

## Abstract

In the last decade, magnetic nanoparticles (MNPs) have attracted much attention for the implementation of non-invasive approaches suitable for the diagnosis and treatment of vascular diseases. In this work, the optimization of novel vascular grafts loaded with Nickel-based nanoparticles via electrospinning is proposed. Two different polycarbonate urethanes—i.e., Corethane A80 (COT) and Chronoflex AL80 (CHF)—were used to fabricate 3D electrospun nanocomposite grafts. SEM analysis showed a homogeneous distribution of fibers, with slight differences in terms of average diameters as a function of the polymer used—(1.14 ± 0.18) µm for COT, and (1.33 ± 0.23) µm for CHF—that tend to disappear in the presence of MNPs—(1.26 ± 0.19) µm and (1.26 ± 0.213) µm for COT/NPs and CHF/NPs, respectively. TGA analyses confirmed the higher ability of CHF to entrap MNPs in the fibers—18.25% with respect to 14.63% for COT—while DSC analyses suggested an effect of MNPs on short-range rearrangements of hard/soft micro-domains of CHF. Accordingly, mechanical tests confirmed a decay of mechanical strength in the presence of MNPs with some differences depending on the matrix—from (6.16 ± 0.33) MPa to (4.55 ± 0.2) MPa (COT), and from (3.67 ± 0.18) MPa to (2.97 ± 0.22) MPa (CNF). The in vitro response revealed that the presence of MNPs did not negatively affect cell viability after 7 days in in vitro culture, suggesting a promising use of these materials as smart vascular grafts able to support the actuation function of vessel wall muscles.

## 1. Introduction

In the presence of vascular occlusion, ischemic phenomena can be extensive, and it is necessary to clinically proceed through the use of vascular grafts able to circumvent ischemic conditions [1]. The use of vascular autografts, allografts and xenografts has represented the optimal clinical solution for facing these problems for many years, despite the concerns associated with uncontrolled autoimmune response and infection risks [2]. Over the last three decades, as technological development in the biomaterials sector has grown, increasing interest has been demonstrated in the design of engineered synthetic grafts capable of replacing natural ones, which are becoming less available due to the scarcity of donors and the rising number of surgical interventions in the vascular field [3].

To this end, several biomaterials have been explored to achieve the required set of chemical/mechanical properties suitable for improving the interface with cells in the luminal region [4]. In view of the direct mechanical pressure supported by the activity of numerous enzymes within the circulation system, the use of low-rate degradable polymers represents a gold standard in the design of vascular engineered structures [5]. Among these, polycarbonate urethanes (PCUs) are certainly the most interesting ones, showing adequate biocompatibility, excellent hemocompatibility and important features in terms of mechanical strength, toughness and resistance to degradation [6,7].

Most interestingly, these properties can be highly tuned by changing the relative weight ratios of hard (isocyanate-based) and soft (polyol-based) segments, which is able to affect flexibility/rigidity and degradation properties as a function of the relative amorphous/crystalline phase content [8,9]. Hence, an extended group of different PU-based vascular grafts has been manufactured by several processing methods based on additive manufacturing techniques (i.e, electrospinning and 3D printing) [10]. In particular, electrospinning is considered an elective technology for the fabrication of vascular grafts, offering the opportunity to develop nanostructured scaffolds with different compositions and 3D architectures by the assembly of multicomponent fibers [11] and single-, double- and triple-layered forms [12].

To date, vascular grafts based on polyurethanes are clinically used to prevent severe vascular injury [13]. However, the repair of vascular structures is a complex process characterized by different stages concerning inflammation phenomena, neo-intimal hyperplasia and vascular remodeling that often cannot be successfully covered by the biomechanical functions of biocompatible grafts [14,15].

In this context, the use of magnetic nanoparticles (MNPs) can offer a relevant strategy for adding new functionalities to fibers, working as an effective and non-invasive means to support vascular repair during different stages, thus providing a suitable tool for both diagnosis and treatment approaches [16]. The incorporation of MNPs into different biomaterials such as hydrogels, membranes and 3D-printed scaffolds has gained attention due to their good biocompatibility and potential applications in the biomedical field [17,18,19]. MNPs have been studied for applications in biosensing, cancer therapy, bone tissue engineering, the delivery of drugs and theranostic applications [20,21]. Several studies have observed the positive influence of MNPs on in vitro cell activities, in terms of adhesion and spatial distribution, with improvements in their functionalities in 3D scaffolds for tissue engineering [22,23]. More recently, the magnetic response of MNP-loaded micro/sub-micro-fibers has been deeply investigated for their potential use as magneto-active systems in the biomedical field [24,25,26]. Based on previous experimental studies, it is herein proposed to investigate MNP-loaded microstructured tubes fabricated via electrospinning techniques by using two different CPUs, namely, Corethane A80 (COT) and Chronoflex AL80 (CHF). A comparative study will be performed to evaluate physicochemical, mechanical and biological properties in order to identify the system suitable for use as a magnetic responsive graft for vascular repair.

## 2. Materials and Methods

### 2.1. Materials

Polycarbonate urethanes (PCUs), i.e., Corethane 80A and Chronoflex 80A, were purchased from AdvanSouce Biomaterials, Wilmington, NC, USA. Nickel nanoparticles (≥99% trace metal basis), Tetrahydrofuran (THF) (≥99.9%, anhydrous, inhibitor-free) and N-N Dimethylformamide (DMAc) (≥99.8%, anhydrous) were purchased from Sigma Aldrich (Milan, Italy).

### 2.2. Electrospinning Process

Corethane 80A (COT) and Chronoflex 80A (CHF) were used for the fabrication of polycarbonate urethane fibers. First, 15% *w*/*v* of polymer was dissolved into a solvent mixture—7:3 (*v*/*v*) Tetrahydrofuran/Dimethylformamide (THF/DMF) ratio—for 6 h until solution homogeneity was completely reached. For magnetic charging, commercial Nickel nanoparticles (MNPs) were used. MNPs were chosen as filler on the basis of the following properties: (i) sizes ranging from 20 to 200 nm (99.99% Aldrich Chemical, Milan, Italy) to assure optimal incorporation in polymer fibers with a micrometric diameter; (ii) capability to preserve considerable magnetization response in comparison to iron oxides [27,28,29]; (iii) high elastomagnetic response when dispersed in a polymer matrix as extensively reported in previous works [24,25,26]. According to previous studies [30], MNPs (12% *v*/*v*) were added after complete dissolution, with the support of manual mixing to improve the dispersion of MNPs in the presence of high-viscosity solution. For the electrospinning process, different formulations were prepared, as reported in Table 1.

Solutions were placed into a 5 mL syringe with an 18 G metal needle and arranged in the electrospinning apparatus (Nanon01, MECC, Fukuoka, Japan). At the first stage, fibers were randomly collected over a grounded aluminum foil target in order to obtain flat membranes. Different process parameters were selected to optimize the final fiber morphology: optimal MNP-loaded fibers were fabricated by setting a needle/electrode distance of 100 mm, a voltage of 18 kV and a flow rate of 1.0 mL/h. The process was carried out in a vertical configuration at 25 °C and 50% relative humidity, and the deposition time was defined to obtain fiber layers suitable for removal from the collector (thickness around 300 μm).

For the preparation of fibrous tubes with a wall thickness of about 1 mm, fibers were collected onto a 1.5 mm diameter metal mandrel for about 30 min, at a rotating rate of 50 rpm (Figure 1), by imposing the same process conditions used for flat membranes. Flat samples with different geometries were obtained as a function of the peculiar characterization tests (i.e., dog bone-shaped for mechanical tests, rounded for cell culture).

### 2.3. Morphological Characterization

The morphology of the fibers was observed by scanning electron microscopy (FESEM, QUANTA200, FEI, Eindhoven, The Netherlands). Prior to the observation, all of the samples were dried in a fume hood for 24 h, mounted on metal stubs and sputter-coated with gold palladium, and they were then analyzed under high-vacuum conditions at low voltage (5 kV) to minimize sample burning under the electron beam. From the SEM images, fiber diameter was measured from selected micrographs using image analysis software (Image J, version 1.39). Fiber mean diameters were calculated from at least 20 measurements from three independent samples.

### 2.4. Physicochemical Analyses

Fourier-transform infrared spectroscopy coupled with the attenuated total reflectance technique (FTIR-ATR—Perkin Elmer Spectrum 100 FTIR spectrophotometer, Milan, Italy) was used to evaluate the chemical compositions of the porous tubes. Spectra were acquired in the spectral region between 500 and 4000 cm^−1^. The analysis was performed using Origin software (OriginPro 8 SR0; OriginLab Corporation, Northampton, MA, USA).

### 2.5. Thermal Analysis

Thermogravimetric analysis (TGA Q500, TA Instrument, Milan, Italy) was carried out under nitrogen flow within a temperature range from 25 to 600 °C and at a scanning rate of 10 °C/min. Weight loss and derivative functions were plotted versus temperature to analyze changes in the peak shape ascribable to the presence of magnetic nanoparticles. Furthermore, the heat exchanged in the phase transitions was determined by a differential scanning calorimeter (DSC, Discovery DSC, TA Instruments, New Castle, DE, USA). Each sample was heated from −50 °C to 100 °C at a rate of 5 °C/min, then cooled from 100 °C to −50 °C at a rate of 5 K/min and heated again from −50 °C to 100 °C. Calorimetric properties were measured in triplicate for each sample, and analysis of variance at a probability level *p* = 0.05—using OriginPro 2018 software (OriginLab Corporation, Northampton, USA)—was performed for assessing statistical significance among the mean values of each calorimetric property.

### 2.6. Mechanical Tests

Dog bone-shaped specimens were obtained through the CEAST C6051 punching machine (Instron, Norwood, MA, USA) equipped with an ASTM D1708 hollow cutting die. Ten specimens for each group of materials were obtained, and they were randomly divided into two samples, each consisting of five specimens, for performing static and dynamic mechanical tests in tensile mode. The width of each specimen was measured using the Mytutoyo CD6 digital calliper (Mytutoyo Corp, Kanagawa, Japan), and the specimens’ thickness was measured through the non-contact laser sensor (Micro-Epsilon optoNCDT 1420-10, MICRO-EPSILON UK & Ireland Ltd., Ortenburg, Germany), while the distance of 22.00 ± 0.05 mm between the titanium lower and upper grips was used as the specimens’ length.

Mechanical static tensile tests at room temperature were performed using the Instron dynamometer 5566 (Instron, Bucks, UK) equipped with a 10 N loading cell (Figure 2a). Tests were carried out according to the standard microtensile test method D1708 at a crosshead speed of 100 mm/min (strain rate = 7.57% s^−1^). A preload of 0.01 N was applied before running the tensile test. Force and elongation data were acquired at a speed of 10 points/s. Stress and strain were obtained by dividing the load and the elongation by the specimen’s cross-section area and the initial length, respectively.

Dynamic mechanical analysis (DMA) in fixed environmental conditions—i.e., T = (24 ± 1) °C and % RH = 50—was performed through the Enduratec Biodynamic dynamometer Bose series (Minnetonka, MN, USA) equipped with a 20 N load cell (Figure 2b). Each test consisted of 3 steps: a monotonic ramp at an elongation speed of 10 mm/s (strain rate = 45.5% s^−1^) up to 10% of strain; a hold step at 10% of strain for 1800 s; and a sinusoidal frequency sweep (mean level and dynamic amplitude of 10% and 1% of strain, respectively) between 0.01 Hz and 10 Hz.

Stress relaxation profiles for each investigated sample were determined from the hold step at 10% of strain for 1800 s, and data were interpolated using the generalized Maxwell model consisting of a spring in parallel with four spring–dashpot elements. This model has already been adopted to describe the viscoelastic behavior of polyurethane fibers [30].

Dynamic mechanical data between 0.01 Hz and 10 Hz were described through the storage and loss moduli (E′ and E″, respectively). E′ and E″ were computed considering the amplitudes and the phase angle between the strain and stress sinusoidal signals.

Analysis of variance at a probability level *p* = 0.05 using OriginPro 2018 software (OriginLab Corporation, Northampton, MA, USA) was performed for assessing statistical significance among the mean values of each investigated mechanical property.

### 2.7. In Vitro Studies

For in vitro biocompatibility studies, human bone marrow mesenchymal stem cells (hBM-MSCs; Sigma-Aldrich, Milan, Italy, SCC034) were used. Before the in vitro studies, circular samples—7 mm in diameter—were sterilized with a solution of 70% ethanol for 15 min, and after that time, samples were rinsed with Phosphate-Buffered Saline (PBS) and dried under the hood.

For adhesion assays, hBM-MSCs were seeded onto COT, COT/NP, CHF and CHF/NP fibers (2 × 10^4^ cells per sample) into a tissue culture plate (TCP) and incubated in Eagle’s alpha minimum essential medium (α-MEM) supplemented with 10% fetal bovine serum, antibiotic solution (100 μg/mL streptomycin and 100 U/mL) and 2 mM L-glutamine. The cell cultures were maintained at 37 °C in a humidified atmosphere with 5% CO_2_ and 95% air over 4 and 24 h. After that period of time, cell adhesion was evaluated by the XTT assay (Roche) based on the cleavage of the yellow tetrazolium salt XTT (Roche Diagnostics Deutshland GmbH, Mannehim, Germany) to form an orange formazan dye by metabolically active cells. The concentration of the formazan product is directly proportional to the number of metabolically active cells. Briefly, samples were washed two times to remove the unattached cells, and a solution with 100 µL of fresh medium, with 50 µL of XTT reagent, was added to the samples and incubated in standard conditions for four hours. After the incubation period, the supernatant was recovered and placed in a 96-well plate reader. Absorbance measurements were recorded at 450 nm with a plate reader (Wallac Victor 1420; PerkinElmer, Boston, MA, USA). The reference wavelength was 650 nm. The results of the cell adhesion are presented as a percentage of adhesion with respect to the tissue culture plate (TCP), following the formula(AS/ATCP) × 100%
where AS corresponds to the absorbance of the sample and ATCP to the absorbance of the TCP.

Cell proliferation was assessed by using cell counting kit-8 (CCK-8, Dojindo, Kumamoto, Japan) to analyze hMSC proliferation at 1, 3, 7 and 14 days. For each experimental time, the culture medium was refreshed and 10% (*v*/*v*) of CCK-8 reagent was added. CCK-8 consists of the conversion of a tetrazolium salt into a soluble formazan dye by the activity of dehydrogenases in viable cells. For the analysis, the medium was removed after 4 h and placed into a 96-well plate reader. To measure the produced formazan, absorbance measurements were recorded at 450 nm in a spectrophotometer (Wallac Victor3 1420, PerkinElmer, Boston, MA, USA). The amount of formazan is directly proportional to the living cells. Results are represented as the mean ± standard deviation and were analyzed by Student’s *t*-tests to determine the differences among the groups, considering *p* < 0.05 as statistically significant.

## 3. Results and Discussion

Polyurethane elastomers represent one of the most relevant classes of segmented copolymers with a broad range of chemical, physical, mechanical [31] and biocompatible [32] properties that make them suitable for biomedical use, i.e., in implants and medical devices. The ability to process them by electrospinning offers the unique opportunity to develop innovative systems with peculiar biomechanical properties and morphological similarity with fibrous components of the extracellular matrix to be successfully used as vascular grafts [33]. In this work, two different polycarbonate urethanes (PCUs)—namely, Corethane 80A (COT) and Chronoflex 80A (CHF)—were proposed for the fabrication of electrospun tubes with elastomagnetic properties, ascribable to the entrapment of magnetic nanoparticles (MNPs).

Figure 3 shows SEM images of COT, CHF, COT/NP and CHF/NP electrospun fibers with homogeneous distribution of randomly arranged smooth fibers, without a significant presence of beads. The average fiber diameter was (1.14 ± 0.18) µm and (1.33 ± 0.23) µm for COT and CHF, respectively. In the case of MNP-loaded fibers, the mean diameter was not changed significantly, being (1.26 ± 0.19) µm and (1.26 ± 0.21) µm for COT/NPs and CHF/NPs, respectively. The SEM images also highlight the presence of small confined agglomerates partially emerging from the fiber surface that confirm the entrapment of MNPs in the polymer matrix.

ATR-FTIR analysis was performed to identify changes in the characteristic peaks of PCUs, due to the presence of MNPs (Figure 4). PCUs show characteristic peaks at 1737 cm^−1^ and 1702 cm^−1^, representing band I of free carbonyls and band II of carbonates, respectively, which are part of the carbonyl region from 1600 to 1730 cm^−1^ [34,35]. No shifts or changes in the peak profiles can be attributed to the presence of MNPs in the case of COT_NP and CHF_NP with respect to the controls—COT and CHF, respectively—confirming the absence of fiber/particle interactions.

Thermogravimetric (TGA) analysis was used to investigate the thermal stability of COT and CHF fibers, and to evaluate their effective ability to entrap MNPs. Figure 5 shows the percentage of mass loss as a function of temperature of all samples.

The thermal degradation depends on the structure of the hard segment and composition of polyurethane [36,37]. In the thermograms, the fibers of COT and CHF displayed an onset degradation temperature of 287 °C and 295 °C, respectively. The temperature at which maximum weight loss occurred was 348 °C for both pure COT and CHF fibers. MNP-loaded fibers showed onset temperatures similar to their counterparts—namely, 284 °C for COT/NPs and 296 °C for CHF/NPs—as well as similar temperatures at which maximum weight loss occurred—366 °C and 354 °C, respectively. More interestingly, a residual weight of 18.25% and 14.63% was recorded in the case of COT/NPs and CHF/NPs, confirming a slight tendency of CHF to better entrap MNPs, related to the intrinsic properties of the polymer used (i.e., viscosity) [38,39].

DSC analyses were performed to evaluate the calorimetric behavior of the nanocomposite fibers. Heat flow profiles related to the first heating (Figure 6a) and second heating stages (Figure 6b) were reported in order to correlate some characteristic temperatures with physical changes, i.e., the amorphous or crystalline state of the materials. In particular the first temperature scan allowed the identification of the characteristic temperature of glass transition of the amorphous soft-segment phase for COT and CHF, equal to −17.4 °C and −21.3 °C, respectively, in agreement with previous studies on bulk systems [40]. The slight significant heat flow difference can be ascribable to the presence of MNPs incorporated into the fibers, as is depicted in the calorimetric measurements reported in Table 2. Figure 6a also allows the identification of some differences related to the presence of different endothermal peaks. In the case of COT fibers, only an endothermic peak (T_2_) is detected at 75.1 °C, and a slight but significant temperature increase is detected in the presence of MNPs. Contrariwise, in the case of CHF, two well-defined peaks can be distinguished, at 50.2 °C and 76.7 °C. It is supposed that they are related to the melting of the crystalline phase of the polymer.

In contrast to the more rigid aromatic segments of COT, the aliphatic hard segments of CHF can be influenced by the presence of MNPs as confirmed by the slight but significant increase in the characteristic temperatures—from 50.2 °C to 50.8 °C and from 76.7 °C to 78.9 °C—in the case of CHF/NPs with respect to the control. This could be explained by short-range rearrangements of hard micro-domains due to the presence of metallic nanoparticles that also influence the chain mobility of adjacent soft-segment micro-domains. This is further confirmed by enthalpy variations (ΔH) recorded in the presence of MNPs, showing a decrease in ΔH_1_—from 0.82 to 0.19 J/g—related to soft-segment micro-domains, perfectly balanced by the increase in ΔH_2_ from 2.10 to 2.67 J/g related to hard-segment ones.

Table 2 reports characteristic temperatures and the enthalpy variation (ΔH) of the endotherms observed through DSC for the investigated materials. Notably, for all the investigated materials, the second temperature scan, occurring after quenching, dramatically differs from the first scan (Figure 6). The endothermic peaks related to the melting of the crystalline phase disappear after quenching from 100 °C to −50 °C. The absence of the melting endotherms during the second temperature scan can be ascribed to the fast cooling process that prevents the polymeric chains from crystallizing [34,41]. However, it is reported that after annealing, even at 30 °C, for a sufficient extended time, an ordered crystalline structure is formed again [42].

Figure 7 reports the static and dynamic micromechanical behavior in tensile mode detected for the investigated samples. Static tensile profiles (Figure 7a) show a starting linear region up to about 15% of strain. Young’s modulus for each investigated sample was computed considering the stress vs. strain slope occurring through this region, and data are reported in Table 3. No significant statistical difference is observed among the Young’s modulus mean values, thus suggesting that both the polymeric matrix and the MNPs do not affect the stiffness of the polyurethane-based matrix.

Similar tensile behavior is observed for each sample up to 50% of strain; however, at higher stretch values, COT-based specimens show a significant stress increase compared to CHF specimens. In particular, the strength (i.e., maximum stress values reported in Table 3) of COT specimens is significantly higher (*p* < 0.05) than those of the other samples, thus suggesting that both the type of polyurethane matrix and the incorporation of iron oxide nanoparticles significantly affect the material’s strength. Moreover, for both polyurethane matrices, nanoparticles significantly reduce (*p* < 0.05) the strength.

The observed static tensile behavior (Figure 7a) for the CHF and COT polyurethanes is similar to the profiles reported in the literature for compact CHF and COT [43,44], suggesting that the static profiles of the investigated electrospun porous polyurethanes are similar to bulk polyurethanes, despite the obvious lower stiffness and strength. The significant difference between CHF and COT at strain levels higher than 50% is ascribed to the soft silicone block polymer providing higher flexibility to CHF. Unfortunately, very little is known about the static behavior of electrospun CHF and COT polyurethanes. However, Todesco et al. has processed CHF through the solvent casting technique [45], obtaining dense membranes whose static behavior is similar to that reported in Figure 6a but characterized by a higher modulus and strength. Similarly, Gradinaru et al. synthesized polyurethanes and porous specimens produced through the solvent casting technique [35], showing tensile profiles similar to that of COT reported in Figure 7a. Moreover, Yeganegi et al. electrospun a synthesized polyurethane incorporating anionic dihydroxyl oligomers for improving cell–material interaction, and as a result, stiffness and strength values slightly lower than those presented in Table 3 were reported [46]. On the other hand, Kang et al. processed Pellethane through electrospinning and obtained tensile strength values slightly higher than those reported in Table 3, while elongation values were significantly lower. The effect of nanoparticles on the static tensile properties of electrospun polyurethanes has been poorly investigated. The results reported in Table 3 suggest that the incorporation of a high amount of iron oxide NPs (i.e., 20% *w*/*w*) in polyurethanes slightly but significantly (*p* < 0.05) reduces strength. Through the use of the solvent casting technique and lower amounts of Fe_3_O_4_ NPs (<2%*w*/*w*), slightly higher strength values have been observed [35]. The Young’s modulus results reported in Table 3 suggest that MNPs do not affect stiffness, and this result is consistent with the findings of Gradinaru et al. [35].

Stress relaxation testing and modeling are considered a central tool for describing the viscoelastic behavior of both synthetic polymers and natural tissues [30,47]. Figure 7b shows an overlapping of the stress relaxation spectra detected for all the investigated samples stretched at a 10% level, suggesting that the type of polyurethane and MNPs do not alter the temporal relaxation spectrum. This result is consistent with the static profiles reported in Figure 7a, showing no significant static difference among the investigated materials up to 50% of strain. The peak stress occurring after the linear ramp at a speed of 10 mm/s is about 0.5 MPa, and no significant difference is observed among the mean values reported in Table 4.

The stress relaxation profiles reported in Figure 7b also show that the stress rapidly decreases in the first stage, reaching an almost plateau level after 1800 s. No significant difference is observed among the plateau stress mean values detected for the investigated samples (Table 3), and a stress decay of about 43% is suggested after 1800 s. The generalized Maxwell model provided an excellent fit to all materials’ temporal relaxation spectra (Chi square < 0.0097 and R^2^ > 0.998), and the relaxation times of the four parallel spring–dashpot elements are reported in Table 4. Comparing the stress relaxation data in Table 4 with the literature data is challenging since stress relaxation largely depends on both the speed at which the viscoelastic material is stretched during the initial linear ramp and the magnitude of the hold signal [30]. Despite it being recognized that stress relaxation testing represents a powerful tool for investigating the behavior of viscoelastic materials [30], very few efforts have been devoted to electrospun polyurethanes and nanocomposites. Xia et al., using compression-molded compact polyurethane-based tensile specimens (length of 35 mm), observed stress relaxation profiles occurring after the elongation rate of 50 mm/min (i.e., strain rate = 2.4% s^−1^) [48]. As a result, a stress decay of 41.6% has been detected for specimens containing the highest amount of the hard segment bis(cyclohexylisocyanate). This stress decay is consistent with but slightly lower than the values reported in Table 3, and differences may be ascribed to the strain rate. Aligned dry-spun thermoplastic polyurethanes have been stretched at a strain rate of 10% s^−1^ by Tan et al., and a stress decay of about 30% has been observed [49]. This decay is significantly lower than the values reported in Table 4, and differences may be attributed to random and aligned fiber architectures.

Figure 7 c,d report the storage and loss moduli (i.e., E′ and E″, respectively) measured over the two decades of frequency sweep. Both E′ and E″ increase as frequency increases, with E″ increasing faster than E′. Table 5 shows E′ and E″ values for the investigated electrospun PU and nanocomposites at fixed frequencies of 0.01 Hz, 0.1 Hz, 1 Hz and 10 Hz. At any fixed frequency, no statistically significant difference is observed among the mean values of E′ and E″.

For all the investigated materials, the ratio between E″ and E′, known as the loss or damping factor, increases from 0.16 to 0.24 as frequency is increased from 0.01 Hz to 10 Hz. This result is consistent with damping factor values reported for PU compact elastomers and carbon black-reinforced PU [50], thus suggesting that the damping of electrospun PU and nanocomposites does not differ from that of a compact PU specimen. Very little is reported in the literature concerning DMA of electrospun PU. Nezarati et al. has investigated the dynamic mechanical behavior of electrospun COT at a frequency of 1 Hz and a strain amplitude of 0.1% over a broad range of temperatures [43]. The reported storage modulus at room temperature is about 1 MPa, lower than the storage modulus value reported for COT in Table 5. This discrepancy may be due to the electrospinning parameters being optimized for obtaining higher compliance matching than that of native vessels [43].

To assess the suitability of electrospun fibers for vascular applications, the biocompatibility of different MNP-loaded fibers was tested in vitro. More specifically, the in vitro response of COT and CHF with and without MNPs was investigated on contact with hBM-MSCs in terms of cell adhesion and cell viability, as reported in Figure 8. Figure 8a shows cell adhesion close to 80% with respect to the control, after 4 and 24 h, in the case of COT, COT/NP, CHF and CHF/NP fibers. Cell interaction was confirmed by the qualitative evaluation of cell morphology after 24 h by SEM (Figure 8c–f). In particular, the spread morphology of cells along the fibers independently of the MNPs’ presence was ascribed to the peculiar morphological signals exerted by electrospun fibers with micrometric diameters—i.e., high surface-to-volume ratio—thus confirming the ability of fibers to support cell adhesion mechanisms, without negligible effects of MNPs.

The influence of MNPs on cell response was further investigated via viability tests as reported in Figure 8b. At day 1, a relevant decrease in cell viability was recorded in the case of COT/NPs with respect to the control COT, ascribable to the potential interaction of cells with MNPs released during the first in vitro culture steps. Different behavior can be identified in the case of CHF and CHF/NPs, which show a comparable response in terms of cell viability. This could be related to the effect of aliphatic segments of CHF able to promote a more efficient arrangement of MNPs in the fibers. Accordingly, MNPs tend to be more strongly entrapped in CHF fibers—as reported by TGA and DSC data—thus limiting the detachment of MNPs and their release in vitro, in agreement with previous studies correlating polyurethane composition and water permeability/degradation [51]. Notably, this effect tends to be attenuated for longer times. Indeed, a slight decrease in cell viability was recorded after three days in both cases, probably due to the cells’ interaction with released MNPs, while no significant differences in cell viability were recorded after 7 days among the different groups, thus confirming the good biological stability of the MNP-loaded fibers in vitro.

In the past, several reports have confirmed the good biocompatibility of scaffolds reinforced with MNPs, mainly used for bone regeneration [52], indicating that the MNPs do not induce a cytotoxic response of hBM-MSCs [53,54]. More recently, strong efforts are addressing the design of vascular grafts with magnetic properties mainly for clinical practice (i.e., coronary artery bypass). Indeed, they can reduce ischemic time during anastomosis, increasing patency rates, by the use of minimally invasive procedures that promote rapid and uniform cell endothelialization [55,56]. According to our previous studies, the peculiar elastomagnetic properties of the proposed tubes could be successfully used to fabricate tubular grafts able to deform under the application of an external magnetic field. Under the application of oscillating contractions via a magnetic induction field (i.e., amplitude 10 mT, low frequencies <2 Hz), the deformation of the grafts is large—with the reduction to 43% in the tube diameter under rest reversible and repeatable in the elastic regime [57], as required for muscles of vessel walls [58]. Accordingly, magnetically charged electrospun tubes may also guarantee the preservation of the typical conditions of blood perfusion [59], being able to not only withstand but also actively support flow rates similar to those typically present in native vessels, by the local application of external forces such as electro-magnetic ones [57].

## 4. Conclusions

In this work, two different polycarbonate urethanes with recognized properties for vascular applications were processed via electrospinning in combination with Ni-based nanoparticles for the fabrication of microstructured grafts with elastomagnetic properties. The effect of MNPs was investigated in terms of thermal and mechanical properties and biocompatibility. The moderate decay in mechanical strength, accompanied by non-significant variations in the elastic modulus of the fibers when magnetic particles were added, confirms the ability of the magnetically charged tubes to be potentially used as implantable devices for in vivo applications.

In this context, several challenges still persist before the use of current devices can be transferred to clinical surgery. The main concerns to be addressed include the need to reach a right balance in terms of biocompatibility and biomechanical properties, also taking into account adapting these devices to the physiological conditions present after the implant. Recent studies reported several failures of small blood vessel devices based on synthetic polymers in terms of endothelialization, adhesion and biodegradability [60]. Otherwise, other studies have confirmed the intrinsic limitation of natural polymers with intrinsic biocompatibility to completely satisfy mechanical performance requirements [61]. From this perspective, the use of polymer blends—i.e., polyurethanes with the addition of ECM-derived proteins [62]—and the considerable versatility of electro-fluid dynamic processes in the fabrication of multi-layered and/or multicomponent tubes with tailored cues [11,63,64] could offer a promising alternative to achieve more favorable biocompatibility profiles [65] without compromising the mechanical performance of the vascular graft.

## Figures and Tables

**Figure 1 polymers-17-01877-f001:**
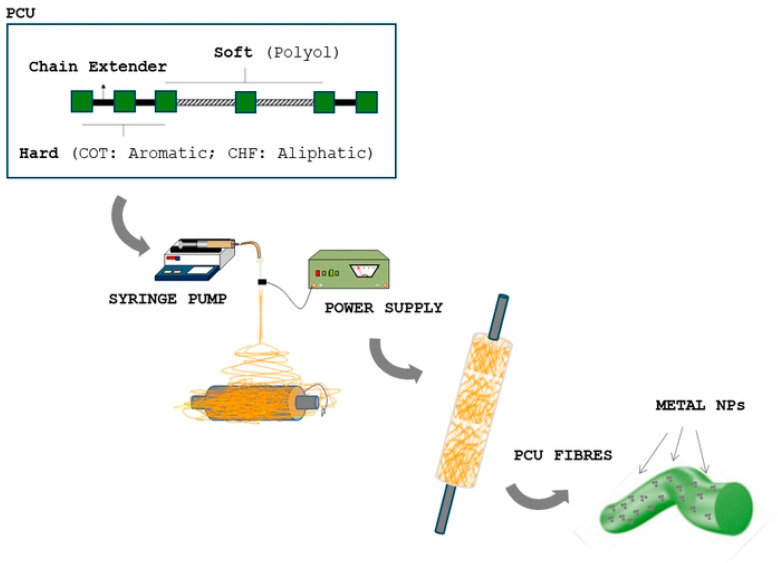
Scheme of preparation of MNP-loaded PCU electrospun tubes.

**Figure 2 polymers-17-01877-f002:**
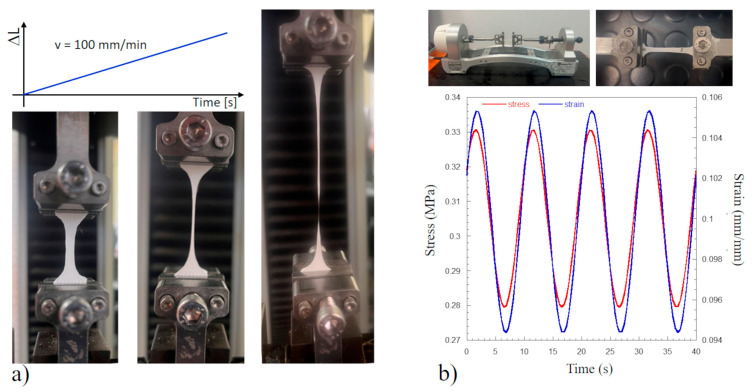
Tensile microtensile tests according to the method D1708: (**a**) static test at a crosshead speed of 100 mm/min showing the typical specimen elongation for CHF; (**b**) dynamic mechanical set-up showing a COT specimen stretched up to 10% of strain and the typical stress response at 0.1 Hz occurring after a hold step for 1800 s.

**Figure 3 polymers-17-01877-f003:**
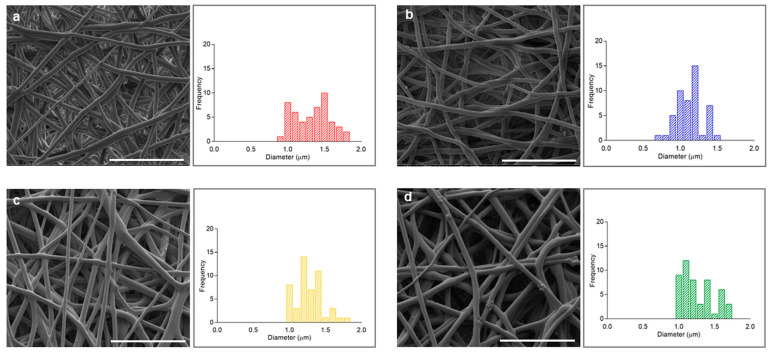
SEM images and fiber diameter distribution of COT (**a**), CHF (**b**), COT/NP (**c**) and CHF/NP (**d**) electrospun tubes (Scale bar: 20 µm).

**Figure 4 polymers-17-01877-f004:**
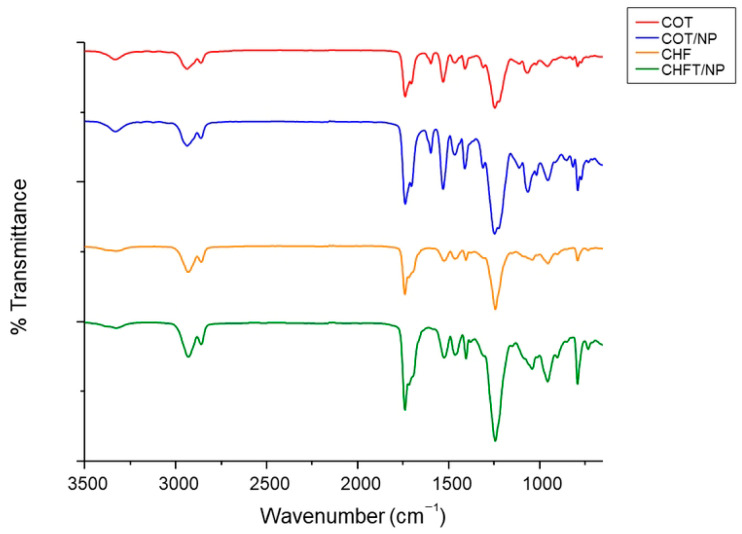
Comparison of ATR-FTIR spectra of COT, COT/NP, CHF and CHF/NP electrospun fibers.

**Figure 5 polymers-17-01877-f005:**
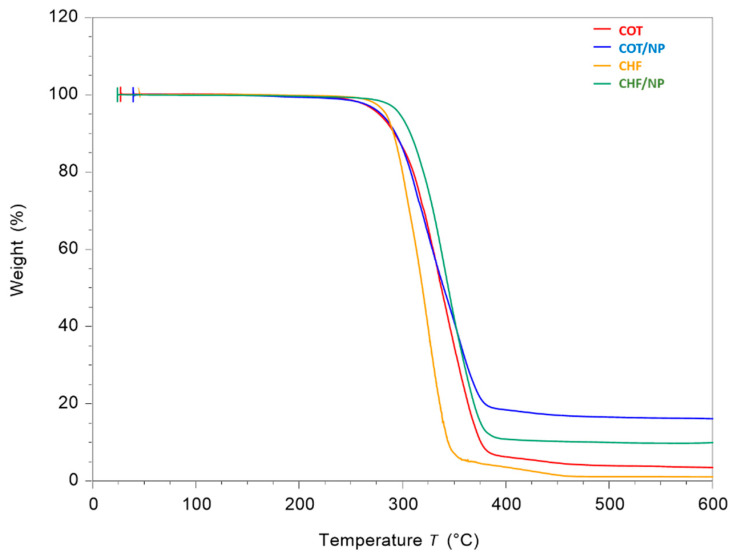
Thermograms of COT, COT/NP, CHF and CHF/NP fibers: evaluation of the weight loss of the fibers as a function of temperature and residue estimation at 600 °C.

**Figure 6 polymers-17-01877-f006:**
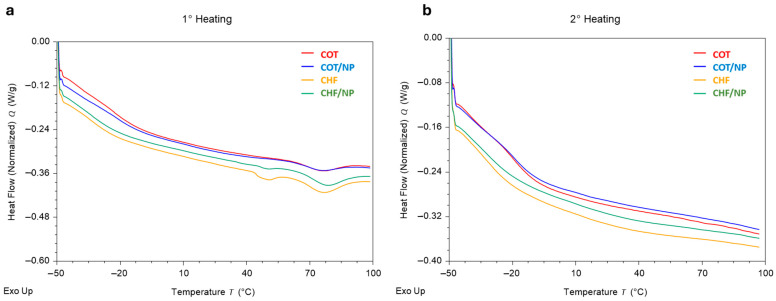
DSC analysis of COT, COT/NP, CHF and CHF/NP fibers. Thermograms after first (**a**) and second (**b**) heating from −50 °C to 100 °C.

**Figure 7 polymers-17-01877-f007:**
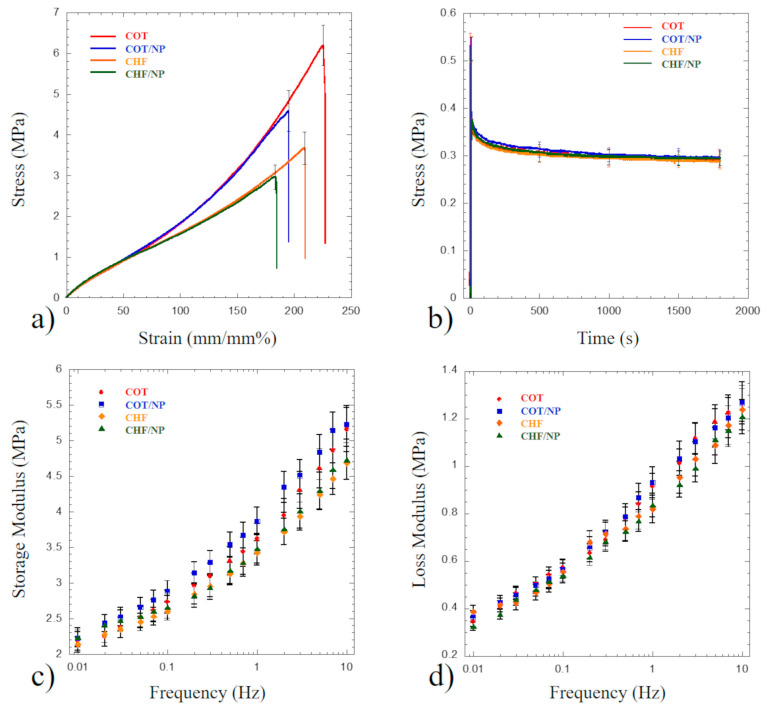
Mechanical behavior of the investigated samples according to ASTM 1708. (**a**) Typical static stress vs. strain tensile profiles; (**b**) typical stress relaxation curves for specimens stretched at 10% of strain; (**c**) storage modulus measured over a three-decade frequency sweep; (**d**) loss modulus measured over a three-decade frequency sweep. The error bars reported in each graph represent the standard deviation.

**Figure 8 polymers-17-01877-f008:**
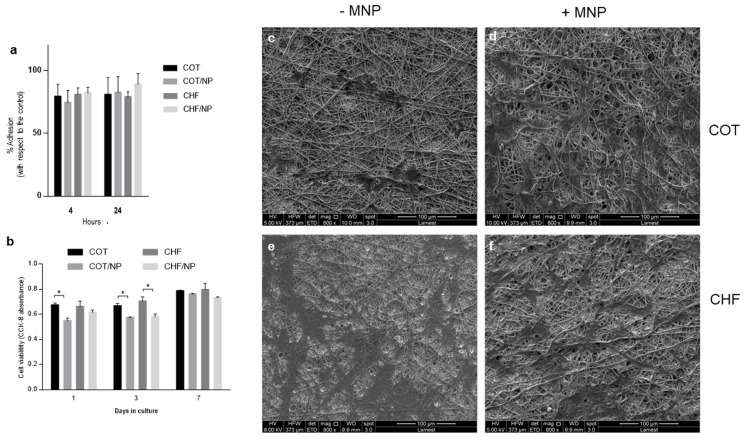
In vitro studies on COT, COT/NP, CHF and CHF/NP fibers: (**a**) cell adhesion; (**b**) cell viability; (**c**–**f**) cell morphology at 24 h. *(*p* < 0.01).

**Table 1 polymers-17-01877-t001:** Summary of fiber composition.

	Polymer	Solvent	MNPs
COT	Corethane (15% *w*/*w*)	THF/DMF 7:3	
COT/NPs	Corethane (15% *w*/*w*)	THF/DMF 7:3	12% *v*/*v*
CHF	Chronoflex (15% *w*/*w*)	THF/DMF 7:3	
CHF/NPs	Chronoflex (15% *w*/*w*)	THF/DMF 7:3	12% *v*/*v*

**Table 2 polymers-17-01877-t002:** Summary of DSC data for amorphous and crystalline phases: glass transition temperatures, melting temperatures and related enthalpy variations (ΔH). Different superscript letters reported in each row of values denote statistically significant differences among the means.

	COT	COT/NPs	CHF	CHF/NPs
	First temperature scan			
T_g_ (°C)	−17.4 ± 0.4 ^a^	−16.3 ± 0.3 ^b^	−21.3 ± 0.4 ^c^	−20.8 ± 0.2 ^d^
ΔH_1_ (J/g)	-	-	0.82 ± 0.03 ^a^	0.19 ± 0.02 ^b^
T_1_ (°C)	-	-	50.2 ± 0.2 ^a^	50.8 ± 0.2 ^b^
ΔH_2_ (J/g)	1.73 ± 0.15 ^a^	1.23 ± 0.16 ^b^	2.10 ± 0.19 ^c^	2.67 ± 0.15 ^d^
T_2_ (°C)	75.1 ± 0.2 ^a^	75.5 ± 0.2 ^b^	76.7 ± 0.3 ^c^	78.9 ± 0.2 ^d^
	Second temperature scan			
T_g_ (°C)	−12.4 ± 0.4 ^a^	−13.8 ± 0.3 ^b^	−23.4 ± 0.3 ^c^	−24.8 ± 0.4 ^d^

**Table 3 polymers-17-01877-t003:** Static tensile properties of the investigated samples elongated at a speed of 100 mm/min (strain rate = 7.57% s^−1^). Different superscript letters reported in each row of values denote statistically significant differences among the means.

	COT	COT/NPs	CHF	CHF/NPs
Young’s Modulus (MPa)	2.92 ± 0.15 ^a^	2.93 ± 0.17 ^a^	2.86 ± 0.14 ^a^	2.87 ± 0.15 ^a^
Maximum Stress (MPa)	6.16 ± 0.33 ^a^	4.55 ± 0.25 ^b^	3.67 ± 0.18 ^c^	2.97 ± 0.22 ^d^
Maximum Strain (mm/mm %)	225 ± 21 ^a^	194 ± 20 ^a,b^	209 ± 20 ^a,b^	183 ± 19 ^b^

**Table 4 polymers-17-01877-t004:** Stress relaxation parameters detected through the profiles reported in Figure 7b and relaxation times suggested by the generalized Maxwell model. Different superscript letters reported in each row of values denote statistically significant differences among the means.

	COT	COT/NPs	CHF	CHF/NPs
Peak stress (MPa)	0.512 ± 0.029 ^a^	0.514 ± 0.027 ^a^	0.513 ± 0.026 ^a^	0.515 ± 0.027 ^a^
Plateau stress (MPa)	0.287 ± 0.017 ^a^	0.292 ± 0.018 ^a^	0.293 ± 0.018 ^a^	0.297 ± 0.019 ^a^
Stress decay (%)	43.9	43.2	42.9	42.3
τ_1_ (s)	697.3	701.6	706.4	705.0
τ_2_ (s)	54.1	52.2	52.4	52.8
τ_3_ (s)	6.8	7.1	6.9	6.5
τ_4_ (s)	0.9	0.9	0.8	0.9

**Table 5 polymers-17-01877-t005:** Storage and loss moduli at fixed frequencies of 0.01 Hz, 0.1 Hz, 1 Hz and 10 Hz. Different superscript letters reported in each row of values denote statistically significant differences among the means.

	COT	COT/NPs	CHF	CHF/NPs
E′ (MPa) @ 0.01 Hz	2.18 ± 0.13 ^a^	2.22 ± 0.16 ^a^	2.13 ± 0.15 ^a^	2.23 ± 0.17 ^a^
E″ (MPa) @ 0.01 Hz	0.37 ± 0.03 ^a^	0.34 ± 0.02 ^a^	0.34 ± 0.02 ^a^	0.36 ± 0.02 ^a^
E′ (MPa) @ 0.1 Hz	2.73 ± 0.16 ^a^	2.88 ± 0.19 ^a^	2.60 ± 0.15 ^a^	2.66 ± 0.18 ^a^
E″ (MPa) @ 0.1 Hz	0.55 ± 0.03 ^a^	0.53 ± 0.03 ^a^	0.57 ± 0.04 ^a^	0.56 ± 0.03 ^a^
E′ (MPa) @ 1 Hz	3.62 ± 0.23 ^a^	3.86 ± 0.27 ^a^	3.42 ± 0.21 ^a^	3.48 ± 0.24 ^a^
E″ (MPa) @ 1 Hz	0.82 ± 0.05 ^a^	0.84 ± 0.05 ^a^	0.91 ± 0.06 ^a^	0.93 ± 0.06 ^a^
E′ (MPa) @ 10 Hz	5.15 ± 0.31 ^a^	5.22 ± 0.33 ^a^	4.68 ± 0.28 ^a^	4.72 ± 0.30 ^a^
E″ (MPa) @ 10 Hz	1.23 ± 0.07 ^a^	1.21 ± 0.07 ^a^	1.26 ± 0.08 ^a^	1.27 ± 0.08 ^a^

## Data Availability

The original contributions presented in this study are included in the article. Further inquiries can be directed to the corresponding authors.

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
