# Peer review of "Novel Magnetically Charged Grafts for Vascular Repair: Process Optimization, Mechanical Characterization and In Vitro Validation"

_polymers, 2025, doi:10.3390/polym17131877_

Round 1
Reviewer 1 Report
Comments and Suggestions for Authors@

can be imorived
Author Response
Reviewer 1 comments
- What type of Magnetic particles is used in this study is not mentioned? Why its used in this work and what are the characterisation already done to conform the activity of the nanoparticles used in this work?
Thank you very much for the comment. The current sentence reported at lines 96-98 ("As magnetic charge, commercial Nichel nanoparticles (MNPs), maximum diameter ≤ 200 nm, were used to ensure an efficient incorporation in microsized fibers, preserving a considerable magnetic moment”) was revised and a new sentence was included to better justify the use of Ni nanoparticles, as follows “Ni nanoparticles were chosen as filler on the basis of the following properties: (i) size ranging from 20 to 200 nm (99,99% Aldrich Chemical) to assure the optimal incorporation in polymer fibers having micrometric diameter; (ii) capability to preserve considerable magnetization response also in comparison to iron oxides [27-29] (iii) high elastomagnetic response when dispersed in a polymer matrix as extensively reported in previous works [24-26]”.
- In the topic 2.2 Electrospinning process- commercial Nichel nanoparticles (MNPs) it is not nichel- it is nickel nanoparticles. Correct this
Thank you for your comment. Text has been amended accordingly
- Take FTIR or XPS to conform the binding or functional nature of the taken materials and add that in result
Thank you for your suggestion. New experiments were conducted via ATR to better investigate the interaction of MNPs and polymer matrix. A new figure collecting FTIR spectra was reported (Figure 4) and some sentences have been added in the results and Discussion section to describe the reported results, as follows:
“ATR-FTIR analysis was performed to recognize changes of characteristic peaks of PCUs, due to the presence of MNPs (Figure 4). PCUs show characteristic peaks at 1737 cm-1 and 1702 cm-1, representing band I of free carbonyls and band II of carbonates respectively, which are part of the carbonyl region from 1600 to 1730 cm-1 [34,35]. No shift or changes of the peak profiles can be attributed to the presence of MNPs in the case of COT_NP and CHF_NP respect to the controls – COT and CHF respectively – confirming the absence of fiber/particle interactions”. (for the Figure4, see the attached PDF file)
- The SEM images of In vitro studies are good but for the cell viability tests you can also add fluorescence spectroscopy its also need to explain the cell growth and the count of living and death cell during the process?
Thank you for the observation. SEM images allowed to investigate cell fiber interactions after 24 hours, giving the opportunity to observe the characteristic spread morphology of hBM-MSCs along the fibers. We demonstrated that cells tend to spread onto the fibers independently of MNPs presence, due to the peculiar morphological signal of the fibrous network, able to support cell adhesion.
Despite fluorescence spectroscopy could offer further information about cell growth, the reliability of this approach could be negligible due to the self-autofluorescence degree of MNPs which can interfere – in some extent - with the fluorescence spectroscopy. Hence, for biocompatibility, CCK-8 assay was used to assess cell proliferation by measuring the amount of formazan produced by living cells which is proportional to the increase in cell number. In the methodology corresponding to the use of CCK-8 kit, some information was added to improve the description of the method as follows: “CCK-8 consists in the conversion of a tetrazolium salt into a soluble formazan dye by the activity of dehydrogenases in viable cells. For the analysis, the medium was removed after 4 h and placed into a 96-well plate reader. To measure the produced formazan, absorbance measurements were recorded at 450 nm in a spectrophotometer (Wallac Victor3 1420, PerkinElmer, Boston, MA, USA). The amount of formazan is directed proportional to the living cells”.
To give you more qualitative information about the amount of cells adhered onto the samples, we reported below new SEM images with lower magnification that clearly highlight a large number of adhered cells onto the samples, independently upon the presence of MNPs. (Figures are attached in the PDF file)

Reviewer 2 Report
Comments and Suggestions for Authors
The manuscript is generally well-structured and clearly presented, addressing an important area in vascular graft design. However, several revisions are needed to enhance clarity and completeness.
- It is unclear whether the magnetic nanoparticles (MNPs) are visible in the SEM images. If they are not distinguishable, the authors should briefly explain why—for example, due to resolution limits or particle distribution within the matrix.
- Figure 6 (C & D) is difficult to interpret due to low visual clarity. These panels should be improved or the data presented in a clearer format to enhance understanding.
- Finally, the manuscript does not include any magnetic characterization data. Since the inclusion of MNPs is central to the novelty of this work, this omission should be addressed by providing appropriate magnetic property measurements to support the claims.
- Formatting needs checking as a large number of full stops are missing.
- Referring to the MNP changes often in the manuscript, please keep consistent.
- Please carefully check the manuscript for meaning, occasionally meaning is lost, for example, on line 414, the word cushioned does not fit.
Author Response
Reviewer 2:
The manuscript is generally well-structured and clearly presented, addressing an important area in vascular graft design. However, several revisions are needed to enhance clarity and completeness.
It is unclear whether the magnetic nanoparticles (MNPs) are visible in the SEM images. If they are not distinguishable, the authors should briefly explain why—for example, due to resolution limits or particle distribution within the matrix.
Thank you for the comment. MNPs show sizes ranging from 20 to 200 nm as declared by the manufacturer. Taking into consideration the characteristic sizes of PU fibers on micrometric scale, the presence of MNPs is not easy to be detected for the reported image magnification. In the figure 3, it is possible to recognize some clusters of MNPs onto the surface of PU microfibers with micrometric sizes. Further SEM images at higher magnification were reported to better remark their presence.
Figure 6 (C & D) is difficult to interpret due to low visual clarity. These panels should be improved or the data presented in a clearer format to enhance understanding.
Thank you for the comments. Figure 7 C and D (after re-numbering) were revised to improve the visual clarity and data interpretation as requested.
Finally, the manuscript does not include any magnetic characterization data. Since the inclusion of MNPs is central to the novelty of this work, this omission should be addressed by providing appropriate magnetic property measurements to support the claims.
Thank you for this comment. The proposed study aims to validate the use MNPs loaded PCU electrospun tubes as magneto-active devices for vascular repair. For this purpose, we have investigated bio-compatibility, mechanical and thermal response of fibrous matrices – made of two PCU with recognized properties for blood interface - to evaluate the effect of magnetic particles loading. Experiments on magnetic response were not specifically reported in this work because a complete characterization of elasto-magnetic response of PCU electrospun tubes was just performed in our previous works [57]. In order to better address on this topic, more information has been included into the manuscript by adding a new sentence at the end of results and discussion section as follows: “Under the application of oscillating contractions via magnetic induction field (i.e., amplitude 10mT, low frequencies <2 Hz), their deformation is large – with a reduction until 43% of the tube diameter under rest - reversible and repeatable in the elastic regime [57], as required for muscles of vessel walls”.
English revisions:
Formatting needs checking as a large number of full stops are missing.
Checked
Referring to the MNP changes often in the manuscript, please keep consistent.
Revised
Please carefully check the manuscript for meaning, occasionally meaning is lost, for example, on line 414, the word cushioned does not fit.
Revised
Round 2
Reviewer 2 Report
Comments and Suggestions for Authors
The authors have adequately answered all comments.
Author Response
The authors have adequately answered all comments.
Thank you for the positive feedback